# Getting Your LLMs Ready for Reinforcement Learning with Lightweight SFT

**Xinran Li**[1,2]    **Guangda Huzhang**[*]    **Siqi Shen**[3]    **Qing-Guo Chen**[2]    **Zhao Xu**[2]
**Weihua Luo**[2]    **Kaifu Zhang**[2]    **Jun Zhang**[1,*]
[1]The Hong Kong University of Science and Technology
[2]Alibaba International Digital Commerce
[3]Fujian Key Laboratory of Sensing and Computing for Smart Cities,
  School of Informatics, Xiamen University, China
`xinran.li@connect.ust.hk, wscmyjy@gmail.com, siqishen@xmu.edu.cn,`
`{qingguo.cqg, changgong.xz, weihua.luowh, kaifu.zkf}@alibaba-inc.com,`
`eejzhang@ust.hk`

## Abstract

Reinforcement learning (RL) has emerged as a powerful post-training paradigm for large language models (LLMs), yet its effectiveness varies significantly across base models. While incorporating a pre-RL supervised fine-tuning (SFT) phase can enhance RL training, key questions remain: how long should the SFT cold-start phase last, and is the SFT objective truly aligned with the requirements for effective RL preparation? In our analysis of cold-start dynamics, we uncover a key limitation: the SFT checkpoint with the highest evaluation performance often fails to maximize RL potential due to distributional forgetting—a phenomenon where the model drifts excessively away from the base model's distribution even before traditional overfitting occurs. We identify diversity metrics, such as the entropy and self-BLEU, as more reliable early-stopping criteria than the standard performance-based checkpoint selection. Our findings show that SFT checkpoints with peak diversity consistently lead to superior post-RL results. Building on these insights, we introduce Adaptive Early-Stop Loss (AESL), a lightweight and dynamic cold-start method that balances the acquisition of new patterns with the preservation of the base model's distribution. AESL operates at both the token and subsequence levels, providing finer-grained control over the cold-start process. Experimental results on mathematical reasoning benchmarks demonstrate that diversity-based early stopping surpasses traditional performance-based SFT, while AESL further enhances RL preparation. By steering LLMs toward better initialization points for RL, AESL consistently achieves superior final performance compared to existing SFT and cold-start strategies. The code is publicly available at `https://github.com/LXXXXR/AESL`.

## 1 Introduction

Large Language Models (LLMs) (Team et al., 2023; Achiam et al., 2023; Grattafiori et al., 2024; Yang et al., 2025) have demonstrated remarkable capabilities across various domains, including natural language processing, question answering, and planning tasks Mirzadeh et al. (2025); Valmeekam et al. (2023); Huang & Chang (2022). Their abundant prior knowledge enables them to tackle previously unattainable challenges, particularly in fields such as mathematical reasoning and code generation (Ahn et al., 2024; Jiang et al., 2024). The foundational capabilities and extensive knowledge embedded in LLMs unlock a wide range of downstream applications, making them indispensable tools for solving sophisticated real-world problems.

While inherently powerful, LLMs often require post-training to better align their capabilities with downstream tasks. Post-training refines base models by strengthening specific abilities, addressing gaps in reasoning patterns, domain expertise, or human preferences Tie et al. (2025); Kumar

---

[*]Corresponding author.

et al. (2025); Shen et al. (2023). Among various post-training paradigms, reinforcement learning (RL) (Barto, 2021; Hu, 2025; Shao et al., 2024; Zheng et al., 2025; Guo et al., 2025) has recently emerged as a particularly promising approach, offering significant performance improvements for LLMs across downstream tasks. RL's advantages are compelling: it eliminates the need of large-scale curated demonstrations and holds the potential to achieve superhuman performance in tasks where well-defined reward signals are available to guide optimization.

However, despite its promise, RL's theoretical benefits often fail to fully materialize in practice. RL algorithms are notoriously sample-inefficient, requiring extensive exploration to learn complex reasoning patterns from scratch. Moreover, RL's effectiveness heavily depends on the quality and characteristics of the base model—models with limited reasoning capabilities may struggle to discover appropriate reasoning patterns and fail to benefit from RL training. To address these limitations, many post-training pipelines incorporate a supervised fine-tuning (SFT) Lambert et al. (2024); Yang et al. (2025); Guo et al. (2025) cold-start phase that enables models to acquire essential reasoning patterns, such as long chain-of-thought (CoT) (Wei et al., 2022) reasoning, before RL training begins. This approach enhances sample efficiency and overall performance by providing a stronger foundation for subsequent RL optimization.

Despite the widespread adoption of cold-start phases, fundamental questions remain unanswered. *How long should the cold-start phase last to optimally prepare base LLMs for subsequent RL training? More critically, is the standard SFT objective of demonstration imitation well aligned with the goal of RL preparation?* These open questions highlight the need for deeper understanding of effective cold-start design to optimize subsequent RL performance.

In this paper, we address these critical questions by investigating cold-start dynamics and proposing methods to optimize their effectiveness of cold-start SFT for RL preparation. Through an analysis of post-RL performance across various cold-start checkpoints, we reveals a misalignment between standard cold-start objectives and RL preparation goals: the best-performing checkpoint after SFT does not necessarily correspond to the optimal starting point for RL due to distribution forgetting that occurs before traditional overfitting. To address this issue, we introduce improved criteria and methods for cold-start design, ensuring better alignment with RL training requirements. Specifically, our contributions are as follows:

- We demonstrate that diversity metrics, such as entropy and self-BLEU, serve as superior early-stopping criteria compared to evaluation performance. Selecting checkpoints based on diversity consistently yields better RL performance than relying solely on evaluation metrics.
- Building on these insights, we propose Adaptive Early-Stop Loss (AESL), a novel cold-start method that dynamically balances new pattern acquisition with base distribution preservation at both token and subsequence levels, providing fine-grained control over the preparation process.
- Through extensive experiments across different base models (Qwen2.5-7B-Instruct (Yang et al., 2024a) and Qwen2.5-Math-7B (Yang et al., 2024b)) and various cold-start data settings, we demonstrate AESL's superior performance, establishing it as an effective solution for preparing LLMs for successful RL training.

## 2 PRELIMINARIES

The goal of the post-training phase for LLMs is to refine base models by enhancing their knowledge and capabilities in specific domains, such as instruction following and reasoning (Kumar et al., 2025). Modern LLM post-training predominantly employs two complementary paradigms: SFT and RL. SFT relies on high-quality demonstration datasets to align model outputs with desired responses, while RL requires only prompt sets and reward signals from verifiers to optimize response generation through policy learning. The RL paradigm has gained significant attention due to its potential to achieve super-human intelligence.

### 2.1 RL FOR LLM POST-TRAINING

In the reinforcement learning with verifiable reward (RLVR) framework (Lambert et al., 2024), the objective is to optimize the policy (i.e., the LLM) $\pi_\theta(s_t|q, s_{<t})$ to maximize the expected reward from a verifier $R(s_t, q)$, where $s_t$ represents the current token output, $s_{<t}$ represent the first $t-1$

generated token and $q$ denotes the input question. In this work, we adopt the widely-used GRPO algorithm (Shao et al., 2024) for RL post-training, which optimizes the following objective:

$$\mathcal{L}_{\text{GRPO}}(\theta) = \mathbb{E}_{q \sim \mathcal{D}_{\text{RL}}, \{s^i\}_{i=1}^G \sim \pi_{\theta_{\text{old}}}(s|q)} \frac{1}{G} \sum_{i=1}^G \frac{1}{|s^i|} \sum_{t=1}^{|s^i|} \Bigg\{$$

$$\min \left[ \frac{\pi_\theta(s_t^i|q, s_{<t}^i)}{\pi_{\theta_{\text{old}}}(s_t^i|q, s_{<t}^i)} \hat{A}_t^i, \text{clip}\left( \frac{\pi_\theta(s_t^i|q, s_{<t}^i)}{\pi_{\theta_{\text{old}}}(s_t^i|q, s_{<t}^i)}, 1 - \epsilon, 1 + \epsilon \right) \hat{A}_t^i \right] - \beta D_{\text{KL}}[\pi_\theta || \pi_{\text{ref}}] \Bigg\},$$
(1)

where $\mathcal{D}_{\text{RL}}$ is the question set used for RL, $G$ is the group size in GRPO, and $s^i$ denotes a complete rollout sequence. The term $\pi_{\theta_{\text{old}}}$ represents the policy before the update, while $\pi_{\text{ref}}$ denotes the reference policy. The advantage estimate is normalized within the group as $\hat{A}_t^i = \frac{r^i - \text{mean}(r)}{\text{std}(r)}$, where $r = R(s_t, q)$ is the verifier that provides reward signals and $r^i$ is the reward for the $i$-th rollout.

## 2.2 RL COLD-START PHASE VIA SFT

Although RL is highly effective at optimizing task-specific performance, its success is strongly influenced by the capabilities of the base model (as discussed in Section 3.1). Additionally, RL training is sample-inefficient, making it challenging to learn complex reasoning patterns, such as long-CoT, from scratch. To address these challenges, many post-training pipelines (Guo et al., 2025; Lambert et al., 2024) include a cold-start phase using SFT. This phase injects essential reasoning patterns, such as long-CoT, into the base model using a small amount of demonstration data, enabling RL to achieve better sample efficiency and performance. Typically, the cold-start phase employs the cross-entropy (CE) loss as the training objective:

$$\mathcal{L}_{\text{CE}}(\theta) = -\mathbb{E}_{q, s^* \sim \mathcal{D}_{\text{SFT}}} \left[ \log \pi_\theta(s_t^*|q, s_{<t}) \right],$$
(2)

where $\mathcal{D}_{\text{SFT}}$ is the demonstration dataset. The CE loss aligns the model's predictions with the provided demonstrations, enabling it to learn desired reasoning patterns before progressing to RL-based post-training.

## 3 METHODOLOGY

In this section, we examine the post-training process holistically and propose an enhanced criterion for early stopping and an improved loss function for the cold-start phase. These enhancements aim to improve final performance after subsequent RL training. First, in Section 3.1, we analyze how the cold-start phase affects the effectiveness of LLM post-training. Our analysis reveals that the best-performing checkpoint after cold-start, as measured by evaluation metrics, often fails to prepare the model optimally for RL training due to distribution forgetting from the base models. We empirically demonstrate that diversity turning points during cold-start training correspond to superior RL potential, motivating diversity-based early stopping as a more effective criterion. Building on this insight, Section 3.2 introduces our Adaptive Early-Stop Loss (AESL) for the cold-start phase, which adaptively balances preservation of the original distribution with adaptation to new demonstration patterns on a token-by-token basis. Together, these contributions offer improved flexibility and superior performance compared to traditional CE-based approaches.

### 3.1 MOTIVATION: UNDERSTANDING THE COLD-START PHASE DYNAMICS

The cold-start phase is crucial for steering base models toward a promising starting point for RL training, particularly when models lack domain-specific knowledge or reasoning capabilities (e.g., long-CoT patterns). To investigate its role, we first evaluate the RL performance of two base models: Qwen2.5-Math-7B (Yang et al., 2024b) and Qwen2.5-7B-Instruct (Yang et al., 2024a), and compared their post-RL performance.

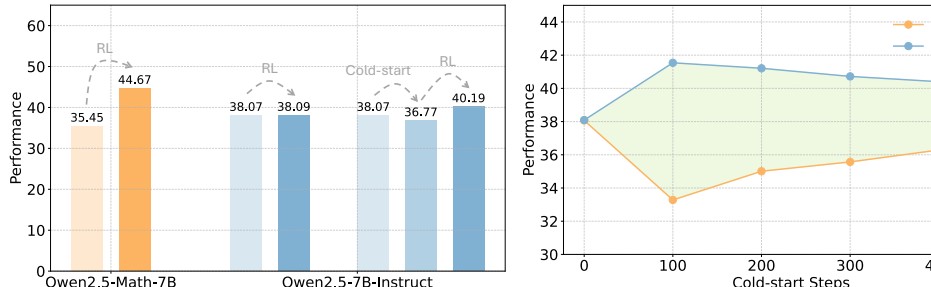

(a) Qwen2.5-Math-7B achieves significant direct performance gains through RL, while Qwen2.5-7B-Instruct requires cold-start preparation for effective subsequent RL training.

(b) Performance before and after RL training across different cold-start training steps. The best post-cold-start checkpoint does not correspond to optimal post-RL performance.

Figure 1: Motivating observations showing the necessity and optimization challenges of cold-start training. Performance represents averages across multiple mathematical reasoning benchmarks (detailed in Section B). Complete results are provided in Section C.1

.

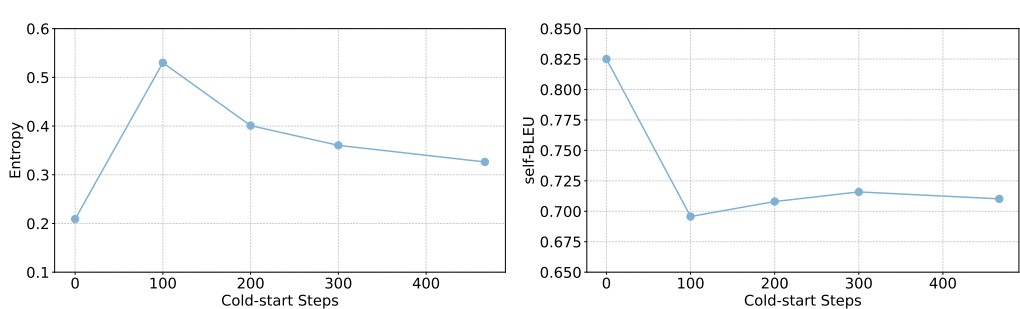

(a) Entropy dynamics during cold-start training. Higher values indicate increased response diversity.

(b) Self-BLEU dynamics during cold-start training. Lower values indicate increased response diversity.

Figure 2: Diversity measurements throughout the cold-start training process corresponding to Figure 1b, revealing the relationship between diversity measurement and RL potential.

**The Necessity of Cold-Start**  As shown in Figure 1a, Qwen2.5-Math-7B achieves significant performance gains directly after RL training. In contrast, Qwen2.5-7B-Instruct shows minimal improvement under direct RL training. This disparity is further reflected in slower growth of average response length (detailed in Section C.1), underscoring the need for a cold-start SFT phase to inject long-CoT patterns into the base model. Following established conventions, we implement a lightweight cold-start phase (using less than $1/10$ of the data volume of the subsequent RL phase) and observe substantially enhanced overall performance gains. This confirms that strategic cold-start SFT can effectively steer base models toward better RL starting points. However, this also raises a critical question: *is optimizing for the best cold-start evaluation performance aligned with achieving the optimal post-RL performance?*

**Misalignment Between Cold-start and RL Objectives**  Our investigation reveals that the objective of the cold-start phase does not necessarily align with the purpose to prepare base model for subsequent RL training, particularly when demonstration data is limited. We examine this by varying training budgets for standard cold-start phases and subsequently applying RL training. Figure 1b presents the performance for both post-cold-start and post-RL models: initial performance decay followed by recovery during cold-start training, representing a shift-and-readaptation process. Crucially, we observe that while evaluation performance continues improving during the readaptation phase, corresponding post-RL performance begins declining. This indicates that the RL potential deteriorates before overfitting to the cold-start dataset, suggesting fundamental misalignment between cold-start objectives (steering models toward better RL starting points) and standard CE loss objectives (maximizing demonstration dataset imitation).

**Diversity as Criteria for Cold-start Early-stopping:** To understand this performance degradation phenomenon, we examine diversity measurements throughout the cold-start training process (Figure 2). This pattern reflects important distribution dynamics during cold-start training. In the early stages, the model balances acquiring new reasoning patterns from the cold-start dataset with retaining knowledge from its original distribution, as illustrated in Figure 3. However, as training progresses, the model begins to over-adapt to the new dataset, leading to distribution forgetting and reduced RL potential. The diversity peak represents an optimal balance point—a "sweet spot" (step 100 in this example) where the model successfully acquires new reasoning patterns while sufficiently retaining distribution from base model for effective RL training. The enhanced entropy at this point likely stems from the model maintaining dual distribution characteristics from both the original base model and the new dataset patterns. Our analysis yields two key insights:

- From an analytical perspective, deterioration of RL potential occurs before SFT overfitting, revealing fundamental misalignment between cold-start objectives (preparing models for RL) and CE loss objectives (maximizing demonstration imitation).
- From a practical perspective, selecting the best-performing post-cold-start checkpoint may be suboptimal for subsequent RL training. Effective cold-start requires balancing new pattern acquisition with original distribution preservation, with diversity measurements serving as valuable indicators of this trade-off.

## 3.2 ADAPTIVE EARLY-STOPPING LOSS

Based on our experimental findings, we reframe the cold-start objective from complete demonstration imitation to achieving an optimal trade-off between learning new patterns and preserving distribution from the original base model, as depicted in Figure 3. The simplest enhancement to traditional cold-start approaches is implementing diversity-based early stopping. Instead of selecting the best-performing checkpoint after SFT cold-start, we select the checkpoint corresponding to the diversity measurement turning point, which indicates the onset of base model distribution forgetting. Based on observations in Figure 1b, this approach should improve subsequent

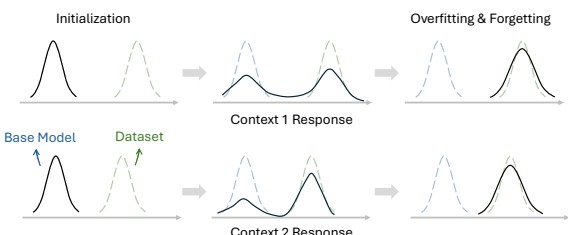

Figure 3: Conceptual illustration of the cold-start training process. The model distribution gradually shifts from the base model toward the cold-start dataset distribution. Overly excessive training leads to overfitting and forgetting of the base model's original distribution, potentially reducing its subsequent RL potential. This highlights the need to balance learning new reasoning patterns from the dataset with preserving the base model's original distribution. Our proposed AESL adaptively addresses this trade-off based on context.

RL post-training performance, which we validate comprehensively in Section 4.

While vanilla early stopping proves effective, it lacks flexibility by applying uniform stopping criteria across the entire dataset. As illustrated in Figure 3, overfitting and distribution forgetting occur at different rates for different tokens and contexts due to varying "distance" between the base model and cold-start dataset distributions. This heterogeneity necessitates a more flexible mechanism for managing the preservation-adaptation trade-off. To address these limitations, we propose the Adaptive Early-Stop Loss (AESL), which provides fine-grained trade-off control at the token and subsequence level rather than dataset-level early stopping. The loss function is a weighted version of the original CE loss:

$$\mathcal{L}_{\text{Ada-stop}}(\theta) = -\mathbb{E}_{q,s^* \sim \mathcal{D}_{\text{SFT}}}\left[p(q, s_t^*, \pi_\theta) \cdot \log \pi_\theta(s_t^*|q, s_{<t})\right], \tag{3}$$

where the adaptive weighting is defined as:

$$p(q, s_t^*, \pi_\theta) = 1 - \text{softmax}\left[\frac{y(s_t^*|q, s_{<t})}{-t_{\text{scaling}} \cdot \frac{1}{|t|}\sum_{i=1}^{t}\log \pi_\theta(s_i^*|q, s_{<i})}\right], \tag{4}$$

with $y$ representing the output logits before softmax normalization, $\text{softmax}[x_i] = \frac{\exp(x_i)}{\sum_j \exp(x_j)}$, and $t_{\text{scaling}}$ being a hyperparameter.

Table 1: Post-training performance (↑) with different cold-start methods. For AIME and AMC, we report avg@64 performance due to smaller test set sizes. For other benchmarks, pass@1 is reported. Avg. denotes the macro-average across benchmarks.

| | AIME24 | AIME25 | AMC23 | MATH. | Min. | Olym. | Avg. |
|---|---|---|---|---|---|---|---|
| Qwen2.5-7B-Instruct | | | | | | | |
| Base | 11.93 | 8.39 | 53.16 | 78.2 | 36.76 | 40.0 | 38.07 |
| +RL | 13.7 | 5.73 | 54.45 | 76.4 | 38.24 | 40.0 | 38.09 |
| +SFT (CE) | 12.29 | 15.31 | 44.22 | 74.2 | 39.34 | 35.26 | 36.77 |
| +RL | 15.99 | 15.52 | 53.36 | 78.4 | 37.13 | 40.74 | 40.19 |
| +SFT (CE-ES) | 9.74 | 12.5 | 43.32 | 71.2 | 31.25 | 31.7 | 33.28 |
| +RL | **18.23** | 15.94 | 54.37 | 80.4 | 37.5 | 42.81 | 41.54 |
| +SFT (GEM) | 13.02 | 14.37 | 45.47 | 74.2 | 34.56 | 33.93 | 35.93 |
| +RL | 16.2 | 15.83 | 53.83 | 80.6 | 36.4 | 40.3 | 40.53 |
| +SFT (PSFT) | 13.33 | 14.69 | 44.73 | 73.6 | 36.03 | 34.52 | 36.15 |
| +RL | 15.73 | 16.15 | 52.15 | 80.6 | **40.07** | 39.11 | 40.64 |
| +SFT (AESL) | 11.41 | 13.65 | 44.26 | 73.8 | 36.4 | 32.44 | 35.33 |
| +RL | 18.18 | **16.88** | **56.48** | **81.8** | 37.13 | **43.11** | **42.26** |
| Qwen2.5-Math-7B | | | | | | | |
| Base | 16.77 | 9.17 | 55.12 | 64.4 | 32.72 | 34.52 | 35.45 |
| +RL | 22.92 | 13.33 | 63.98 | 82.8 | 39.34 | 45.63 | 44.67 |
| +SFT (CE-(ES)) | 20.26 | 19.11 | 59.61 | 83.6 | 40.07 | 41.93 | 44.1 |
| +RL | 24.11 | 21.3 | 69.38 | 86.6 | 41.18 | 49.04 | 48.6 |
| +SFT (GEM) | 17.24 | 20.83 | 58.44 | 81.4 | 41.18 | 43.85 | 43.82 |
| +RL | **26.15** | 21.51 | 68.24 | **87.4** | 41.54 | 51.7 | 49.42 |
| +SFT (PSFT) | 19.32 | 21.46 | 60.08 | 82.4 | 41.18 | 42.37 | 44.47 |
| +RL | 24.27 | **23.12** | 65.74 | 87.0 | 40.44 | 47.56 | 48.02 |
| +SFT (AESL) | 17.29 | 18.75 | 58.16 | 83.6 | 34.93 | 40.3 | 42.17 |
| +RL | 25.0 | 21.98 | **71.25** | 87.0 | **42.28** | **52.74** | **50.04** |

[*] MATH. denotes MATH-500, Min. denotes Minerva Math, and Olym. denotes OlympiadBench.
[*] Gray shading indicates methods for easier readability. Best performance is **bolded**, and second-best is underlined.
[*] For Qwen2.5-Math-7B, CE and CE-ES checkpoints coincide, so results are reported together.

At the token level, the weighting function gradually reduces the loss contribution for tokens where the ground truth already corresponds to high probability under the current policy. This mechanism slows learning when the model already assigns high probabilities to correct tokens, thereby preserving original knowledge while still allowing adaptation to new patterns. AESL also incorporates subsequence-level considerations, recognizing that effective RL preparation requires the ability to generate diverse and plausible reasoning paths. Since sequence diversity measured by entropy $H(s_t|q)$ can be decomposed as: $H(s_t|q) = \sum_{s_{t-1}} \pi(s_{t-1}|q)H(s_t|s_{t-1}) + H(s_{t-1}|q)$ (detailed in Section A), tokens following high-probability prefixes contribute more to overall diversity. Building on this insight, we scale the token-level early stopping with the average log-probability of the prefix context (denominator in Equation (4)). This encourages the model to maintain base distribution when the prefix already aligns closely with the dataset distribution, balancing adaptation to demonstrations with preservation of distribution of the base models.

# 4 EXPERIMENTS

In this section, we integrate the Adaptive Early-Stop Loss (AESL) cold-start method into standard post-training pipelines and evaluate its effectiveness in preparing LLMs for RL post-training. We first compare AESL against baseline cold-start methods, demonstrating its superior performance across multiple base models. We then analyze the impact of dataset configurations and hyperparameters, validating the robustness of AESL in diverse experimental. More experiments are included in Section D.

## 4.1 SETUP

**Base Models** We evaluate our approach using two distinct base models: Qwen2.5-7B-Instruct and Qwen2.5-Math-7B. Both models do not present the ability to reason through long CoT pattern (as demonstrated in Table 13), making the cold-start phase essential for injecting long CoT reasoning patterns. These models represent different starting points for RL post-training: Qwen2.5-7B-Instruct excels at instruction following but may not be optimally positioned for RLVR (as demonstrated in Figure 1a), while Qwen2.5-Math-7B, having been pre-trained on mathematical datasets, generates more diverse responses and serves as a stronger foundation for RLVR. While Qwen2.5-Math-7B requires less intervention during cold-start, we include it to gauge AESL's effectiveness in different initial conditions.

**Dataset and Algorithm** Following established practices (Fu et al., 2025; Zhang et al., 2025), we employ the Openr1-Math-46k-8192 Hugging Face (2025); Yan et al. (2025) dataset for post-training. This dataset comprises 46k long CoT demonstrations generated by the DeepSeek-R1 model. For SFT cold-start, we uniformly subsample 3k examples, ensuring each question corresponds to one correct demonstration verified using Math-Verify[1]. For RL post-training, we utilize the complete 46k question set without demonstrations, employing the GRPO algorithm (Shao et al., 2024).

**Baselines** We compare AESL against five baseline approaches: (i) *Direct RL*: applying RL directly to base models without cold-start preparation; (ii) *CE*: conducting SFT cold-start with CE loss and selecting the best-performing checkpoint for subsequent RL; (iii) *CE-ES (CE with early-stopping)*: conducting SFT cold-start with CE loss and selecting the checkpoint at the diversity turning point for subsequent RL (motivated by findings in Section 3.1); (iv) GEM (Li et al., 2025); and (v) PSFT (Zhu et al., 2025).

Further details about experimental setups and implementation specifics are provided in Section B.

## 4.2 RESULTS

**Main Results** The comparative performance of AESL and baseline methods is summarized in Table 1. Overall, AESL achieves superior performance for both base models, consistently outperforming baselines in subsequent RL training. This highlights AESL's ability to effectively steer LLMs into a better starting point for RLVR. As discussed in Section 3.2, AESL achieves this performance boost by carefully balancing the need to the base model's core distribution while simultaneously learning new reasoning patterns. As shown in Table 2, AESL produces the highest entropy and lowest self-BLEU scores, indicating increased diversity in model outputs after cold-start. This aligns with the findings in Section 3.1, where greater diversity translates to better RL potential. Moreover, BLEU scores between pre- and post-cold-start outputs indicate that AESL maintains a closer alignment with the base model's original output distribution (0.140 vs. 0.135 for CE). This indicates that AESL-trained models retain stronger base knowledge, enabling them to more effectively sample from the base distribution during RL training. Importantly, the CE-ES method also outperforms CE in providing a better starting point for RLVR, despite having inferior performance immediately after the cold-start phase. This observation, coupled with AESL's success, supports the argument in Section 3.2 that the cold-start phase should not solely focus on intimating the demonstration dataset or maximizing evaluation scores but should also consider preserving the base model's existing capabilities, particularly when the dataset size for cold-start is limited. This is further supported by the increased entropy and reduced self-BLEU scores achieved by AESL, as shown in Table 2. Extended discussion is included in Section C.3.

Table 2: Response diversity measurement after cold-start with Qwen2.5-7B-Instruct. Higher entropy and lower self-BLEU indicate greater diversity.

| Method | Entropy($\uparrow$) | Self-BLEU($\downarrow$) |
|--------|---------|-----------|
| CE | 0.326 | 0.710 |
| CE-ES | 0.53 | 0.696 |
| AESL | **0.553** | **0.694** |

**Results with Different Cold-start Datasets** To assess the robustness of AESL, we evaluate its performance under varying cold-start dataset sizes (1k, 3k, and 6k samples) and question

---

[1] https://github.com/huggingface/Math-Verify

Table 3: Performance(↑) comparison across different cold-start dataset sizes using Qwen2.5-7B-Instruct as the base model.

| | AIME24 | AIME25 | AMC23 | MATH. | Min. | Olym. | Avg. |
|---|---|---|---|---|---|---|---|
| Base | 11.93 | 8.39 | 53.16 | 78.2 | 36.76 | 40.0 | 38.07 |
| +RL | 13.7 | 5.73 | 54.45 | 76.4 | 38.24 | 40.0 | 38.09 |
| 1k cold-start dataset | | | | | | | |
| +SFT (CE-(ES)) | 11.67 | 13.44 | 44.92 | 72.8 | 37.5 | 33.33 | 35.61 |
| +RL | 16.2 | 14.74 | 52.62 | **80.8** | 39.71 | 40.74 | 40.8 |
| +SFT (AESL) | 12.4 | 12.24 | 46.13 | 74.0 | 37.13 | 34.52 | 36.07 |
| +RL | **17.14** | **15.36** | **54.02** | 76.8 | **41.18** | **42.37** | **41.14** |
| 6k cold-start dataset | | | | | | | |
| +SFT (CE) | 12.92 | 15.99 | 45.74 | 75.0 | 36.4 | 36.44 | 37.08 |
| +RL | 14.32 | 16.04 | 54.26 | 77.4 | 39.71 | 40.44 | 40.36 |
| +SFT (CE-ES) | 11.61 | 14.22 | 42.97 | 71.6 | 34.56 | 32.15 | 34.52 |
| +RL | 16.56 | 16.72 | 53.79 | 80.2 | 41.54 | **42.37** | 41.86 |
| +SFT (AESL) | 12.55 | 15.73 | 44.84 | 73.6 | 34.19 | 33.63 | 35.76 |
| +RL | **16.67** | **16.82** | **56.41** | **81.6** | **43.01** | 42.22 | **42.79** |

[*] For 1k dataset size, CE and CE-ES checkpoints coincide, so results are reported together.

Table 4: Performance(↑) comparison across different cold-start dataset difficulty splits using Qwen2.5-7B-Instruct as the base model.

| | AIME24 | AIME25 | AMC23 | MATH. | Min. | Olym. | Avg. |
|---|---|---|---|---|---|---|---|
| Base | 11.93 | 8.39 | 53.16 | 78.2 | 36.76 | 40.0 | 38.07 |
| +RL | 13.7 | 5.73 | 54.45 | 76.4 | 38.24 | 40.0 | 38.09 |
| 3k dataset with hard questions (filtered with the base model) | | | | | | | |
| +SFT (CE) | 13.49 | 14.95 | 46.48 | 74.0 | 36.4 | 36.15 | 36.91 |
| +RL | 16.93 | 16.35 | 53.91 | 79.2 | 38.24 | 39.26 | 40.65 |
| +SFT (CE-ES) | 12.81 | 13.91 | 46.48 | 74.0 | 36.76 | 34.22 | 36.36 |
| +RL | 17.03 | 15.78 | **54.14** | 79.4 | 38.24 | 40.0 | 40.77 |
| +SFT (AESL) | 13.7 | 16.04 | 46.56 | 75.2 | 33.46 | 33.48 | 36.41 |
| +RL | **17.55** | **17.03** | **54.14** | **80.4** | **40.07** | **40.89** | **41.68** |
| 3k dataset with simple questions (filtered with the base model) | | | | | | | |
| +SFT (CE) | 12.19 | 13.33 | 44.69 | 72.4 | 35.66 | 34.81 | 35.51 |
| +RL | 14.43 | 15.42 | 53.91 | 80.6 | 40.81 | 40.74 | 40.99 |
| +SFT (CE-ES) | 11.15 | 12.03 | 44.65 | 71.4 | 33.46 | 33.19 | 34.31 |
| +RL | 14.9 | 15.78 | 51.88 | 77.0 | 37.13 | 39.41 | 39.35 |
| +SFT (AESL) | 10.78 | 12.03 | 45.47 | 73.2 | 33.09 | 33.48 | 34.68 |
| +RL | **16.77** | **16.46** | **55.35** | **80.8** | **38.97** | **45.04** | **42.23** |

difficulty levels (simple and hard splits). The results, shown in Table 3 and Table 4, demonstrate that AESL consistently outperforms traditional CE-based cold-start methods across all dataset sizes and difficulty levels. These findings highlight AESL's versatility and effectiveness in diverse post-training scenarios, even under varying data availability constraints. Additional discussions are provided in Section C.3.

**Results with Different Model Families** To further validate AESL's effectiveness, we extended the method to post-train the Llama-3.1-8B-Instruct model (Grattafiori et al., 2024). The results, detailed in Table 5, confirm AESL's ability to enhance post-training performance across different base models. This demonstrates that the proposed approach is not only effective for Qwen-based models but also generalizes well to other architectures.

Table 5: Post-training performance (↑) with different cold-start methods using Llama-3.1-8B-Instruct as the base model.

|  | AIME24 | AIME25 | AMC23 | MATH. | Min. | Olym. | Avg. |
|---|---|---|---|---|---|---|---|
| Base | 5.26 | 0.68 | 19.14 | 45.6 | 23.16 | 16.89 | 18.46 |
| +RL | 4.22 | 1.09 | 25.78 | 50.6 | 27.57 | 15.41 | 20.78 |
| +SFT (CE) | 1.82 | 2.66 | 24.57 | 51.6 | 25.0 | 18.81 | 20.74 |
| +RL | 4.17 | 2.45 | 24.57 | 58.0 | 24.26 | 21.33 | 22.46 |
| +SFT (CE-ES) | 1.88 | 0.68 | 22.7 | 50.6 | 22.79 | 17.78 | 19.4 |
| +RL | 5.36 | 1.93 | 28.12 | 59.8 | **29.04** | 22.22 | 24.41 |
| +SFT (AESL) | 2.6 | 1.61 | 20.43 | 47.8 | 20.22 | 15.85 | 18.08 |
| +RL | **6.56** | **3.8** | **31.21** | **60.4** | 28.68 | **22.37** | **25.5** |

Table 6: Ablation studies for AESL cold-start using Qwen2.5-7B-Instruct as the base model. Results indicate post-RL performance.

|  | AIME24 | AIME25 | AMC23 | MATH. | Min. | Olym. | Avg. |
|---|---|---|---|---|---|---|---|
| Base | 13.7 | 5.73 | 54.45 | 76.4 | 38.24 | 40.0 | 38.09 |
| AESL | **18.18** | **16.88** | **56.48** | **81.8** | 37.13 | **43.11** | **42.26** |
| ASEL w/ const. d. | 16.51 | 15.0 | 52.62 | 78.2 | 40.07 | 39.7 | 40.35 |
| ASEL w/ rank | 14.58 | 14.95 | 54.18 | 78.6 | 39.34 | 41.93 | 40.6 |
| ASEL w/ linear w. | 16.09 | 15.62 | 52.54 | 79.6 | **41.18** | 40.74 | 40.96 |

**Ablations and Hyperparameter Analysis** We conduct ablation studies to evaluate the impact of key components in AESL, with the results presented in Table 6. Specifically, we compare the full AESL method with four ablations: 1) *ASEL w/ const. d.*: In this variant, the denominator in Equation (4) is replaced with a constant, removing the dynamic adjustment based on prefix prediction confidence. 2) *ASEL w/ rank*: In this variant, we mask the token loss when the ground truth token has the highest predicted probability (Top-1), effectively ignoring such tokens. 3) *ASEL w/ linear w.*: In this version, the token-level SFT loss is weighted inversely proportional to the prefix prediction confidence. The results indicate performance drops across all ablations, underscoring the importance of explicitly accounting for prefix prediction confidence, as implemented in AESL. This finding aligns with the motivation discussed in Section A, demonstrating that the absence of prefix-aware adjustments in the ablations leads to less effective cold-start.

We then examine the impact of the temperature scaling hyperparameter $t_{\text{scaling}}$, which controls the balance between learning new reasoning patterns and preserving distribution from base model. Table 7 reveals that extreme values degrade performance: excessively small $t_{\text{scaling}}$ prevents effective pattern learning, while excessively large values cause base distribution forgetting and reduced diversity. However, within the optimal range (3-5), AESL consistently delivers performance improvements over all baseline methods, demonstrating robust performance across reasonable hyperparameter settings.

## 5 RELATED WORKS

The pretraining stage of LLMs primarily focuses on constructing a general foundation of knowledge and capabilities, whereas the post-training stage adapts these models to specific tasks and requirements (Tie et al., 2025). Two widely used methodologies for post-training are: SFT Guo et al. (2025); Lambert et al. (2024) and RL (Shao et al., 2024; Hu, 2025; Yu et al., 2025; Zheng et al., 2025; Hu et al., 2025), which are employed to inject domain knowledge or enhance specific abilities in LLMs. Recent studies have explored the strengths and limitations of these two paradigms. For instance, Sun et al. (2025) and Muennighoff et al. (2025) analyzed SFT datasets, highlighting the importance of data curation and scaling in improving model performance. In contrast, Yue et al. (2025) argued that RLVR does not introduce new reasoning patterns but is instead constrained by the base model's underlying capabilities. Bridging these observations, Chu et al. (2025) demonstrated that while RL improves generalization, SFT tends to memorize training data but helps stabilize output formats for LLMs.

Table 7: Hyperparameter analysis for $t_{\text{scaling}}$ in AESL using Qwen2.5-7B-Instruct as the base model.

| | AIME24 | AIME25 | AMC23 | MATH. | Min. | Olym. | Avg. |
|---|---|---|---|---|---|---|---|
| Base | 11.93 | 8.39 | 53.16 | 78.2 | 36.76 | 40.0 | 38.07 |
| +RL | 13.7 | 5.73 | 54.45 | 76.4 | 38.24 | 40.0 | 38.09 |
| +SFT ($t_{\text{scaling}} = 1$) | 9.43 | 12.4 | 43.05 | 70.6 | 31.99 | 32.44 | 33.32 |
| +RL | 17.45 | 15.21 | 52.7 | 79.8 | 38.24 | 43.11 | 41.09 |
| +SFT ($t_{\text{scaling}} = 3$) | 12.34 | 14.27 | 44.14 | 72.8 | 36.76 | 33.63 | 35.66 |
| +RL | 18.23 | 15.57 | 55.78 | 80.4 | 38.97 | 44.89 | **42.31** |
| +SFT ($t_{\text{scaling}} = 5$) | 11.41 | 13.65 | 44.26 | 73.38 | 36.4 | 32.44 | 35.33 |
| +RL | 12.18 | 16.88 | 56.48 | 81.8 | 37.13 | 43.11 | 42.26 |
| +SFT ($t_{\text{scaling}} = 7$) | 13.28 | 13.54 | 45.98 | 72.8 | 36.03 | 31.85 | 35.58 |
| +RL | 18.44 | 14.84 | 55.16 | 79.4 | 37.87 | 40.89 | 41.1 |

Building on this foundational understanding of SFT and RL in post-training, recent research has proposed methods to combine the two paradigms more effectively. One promising avenue involves enhancing SFT through evolving data strategies, where data progressively guides the base model during training. For example, Light-R1 (Wen et al., 2025) employed curriculum learning to gradually increase the difficulty of SFT question sets, thereby improving the learning landscape. Similarly, REFT (Luong et al., 2024) leveraged multiple sampling of model outputs, selecting the best trace for SFT instead of relying on single demonstration paths. Another line of research integrates SFT with RL in a combined training stage, allowing demonstrations to better guide RL exploration. LUFFY (Yan et al., 2025) approached this from a data perspective, mixing off-policy demonstrations with on-policy rollouts to improve RL performance. SRFT (Fu et al., 2025) and CHORD (Zhang et al., 2025) tackled the problem through modified objectives, reframing SFT as an auxiliary task within the RL framework.

Our work, alongside GEM (Li et al., 2025) and the concurrent work PSFT (Zhu et al., 2025) , belongs to a third category of approaches that aim to refine the SFT stage by shifting the training objective beyond simple dataset imitation. GEM emphasizes promoting diversity, while PSFT focuses on out-of-distribution generalization using abundant SFT datasets. In contrast, our proposed AESL targets a lightweight cold-start SFT setting (using less than 1/10th the data size of RL training). AESL addresses the challenge of preparing the base model for subsequent RLVR by striking a better balance between learning new reasoning patterns and preserving the base model's prior knowledge.

## 6 CONCLUSIONS

In this work, we addressed the critical challenge of preparing LLMs for RL training through effective cold-start SFT. Our analysis uncovered a fundamental misalignment between traditional SFT objectives and RL preparation goals: the best-performing cold-start checkpoint, as measured by evaluation performance, fails to maximize RL potential. This failure arises due to distribution forgetting from the base models, which occurs before traditional overfitting. We demonstrated that diversity metrics, such as entropy and self-BLEU, are more reliable early-stopping criteria, with peak diversity checkpoints consistently yielding superior post-RL results. Building on this insight, we introduced Adaptive Early-Stop Loss (AESL), a novel cold-start method that dynamically balances new pattern acquisition with the preservation of the base model's distribution at both the token and subsequence levels. Experimental results across mathematical reasoning benchmarks demonstrate that AESL consistently outperforms traditional cold-start methods and competitive baselines, achieving superior final performance in the lightweight SFT setting.

While our work addresses cold-start optimization from an objective perspective, several important aspects warrant further investigation. From an evaluation standpoint, developing methods to predict the upper bound of subsequent RL training could enable more informed cold-start design decisions. From a data curation perspective, understanding how to optimally select and structure cold-start datasets to maximize RL preparation effectiveness remains an open challenge. These directions represent promising avenues for future research in LLM post-training.

## REPRODUCIBILITY STATEMENT

We provide comprehensive details on prompting strategies, datasets, hyperparameters used for training and evaluation, as well as the infrastructure of our experimental setups in Section B. The code is publicly available at `https://github.com/LXXXXR/AESL`.

## ACKNOWLEDGMENT

This work was supported by the Hong Kong Research Grants Council under the Areas of Excellence scheme grant AoE/E-601/22-R and NSFC/RGC Collaborative Research Scheme grant CRS_HKUST603/22. We thank the anonymous reviewers for their valuable feedback and suggestions.

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

## A   DETAILS ON DIVERSITY ANALYSIS

Following previous work (Fu et al., 2025), we formulate the LLM's token generation process as a Markov Decision Process (MDP). The state $s_t$ is defined as the concatenation of all tokens generated so far, making the transition deterministic in this MDP. As discussed in Section 3.1, diversity serves as an indicator of the balance between learning new reasoning patterns and preserving base distribution. To analyze this, we examine the entropy at the sequence level and its relationship to token-level entropy, which corresponds to next-token prediction using cross-entropy (CE) loss:

$$H(s_t|q) = -\sum_{s_t} \pi(s_t|q) \log \pi(s_t|q) \tag{5}$$

$$= -\sum_{s_t} \pi(s_t|s_{t-1})\pi(s_{t-1}|q) \log \left[\pi(s_t|s_{t-1})\pi(s_{t-1}|q)\right] \tag{6}$$

$$= -\sum_{s_t} \pi(s_t|s_{t-1})\pi(s_{t-1}|q) \log \pi(s_t|s_{t-1}) - \sum_{s_t} \pi(s_t|s_{t-1})\pi(s_{t-1}|q) \log \pi(s_{t-1}|q) \tag{7}$$

$$= \sum_{s_{t-1}} \pi(s_{t-1}|q)H(s_t|s_{t-1}) + H(s_{t-1}|q). \tag{8}$$

In online RL, where the data distribution is induced by the policy $\pi$, preserving token-level diversity is equivalent to preserving sequence-level diversity. During the SFT cold-start stage, entropy under different contexts contributes differently to overall sequence diversity. Therefore, we leverage this insight to adaptively adjust the balance between learning new reasoning patterns and preserving distribution from base model, based on prefix prediction accuracy.

## B   DETAILS ON EXPERIMENTAL SETUPS

### B.1   PROMPTING

Following established conventions (Guo et al., 2025), we append the following context to math problems to prompt CoT reasoning:

> Please reason step by step, and put your final answer within \\boxed{}.

Additionally, we enforce that the model starts its response with `<think>\n` to ensure reasoning is explicitly triggered during both training and evaluation.

### B.2   TRAINING

We use the OpenRHLF framework[2] (Hu et al., 2024) to conduct all the experiments with hyper-parameters listed in Table 8 and Table 9. For AESL-specific hyperparameter, we set $t_{\text{scaling}} = 5$. For baseline GEM[3] (Li et al., 2025) and PSFT[4] (Zhu et al., 2025), we use the respective official codebase with recommended hyperparameters. For RL reward function, we use the outcome based reward with 1 for correct answers and 0 for incorrect ones.

Given the maximum sequence length for Qwen2.5-Math-7B is 4096, we increase the RoPE theta from 10,000 to 40,000 and expand the window size to 16,384 to support longer outputs.

### B.3   EVALUATION

In this work, we use the evaluation framework [5] proposed by Liu et al. (2025), with test-time parameters listed in Table 10.

---

[2] https://github.com/OpenRLHF/OpenRLHF.
[3] https://github.com/liziniu/GEM.
[4] https://github.com/zwhong714/PSFT.
[5] https://github.com/sail-sg/understand-r1-zero.

Table 8: Common hyperparameters used for AESL and baselines during SFT cold-start.

| Hyperparameter | Value |
|---|---|
| Learning rate | $5 \times 10^{-6}$ |
| Learning rate scheduler | cosine with minimal lr |
| $\beta_1$, $\beta_2$ | 0.9, 0.95 |
| Learning rate warm up ratio | 0.1 |
| Weight decay | 0.1 |
| Batch size | 64 |
| Number of epochs | 10 |

Table 9: Common hyperparameters used for AESL and baselines during RL phase.

| Hyperparameter | Value |
|---|---|
| Learning rate | $1 \times 10^{-6}$ |
| Learning rate scheduler | cosine with minimal lr |
| $\beta_1$, $\beta_2$ | 0.9, 0.95 |
| Learning rate warm up ratio | 0.1 |
| KL coefficient | 0.001 |
| Batch size | 128 |
| Number of samples per prompt | 8 |
| Number of episode | 1 |
| Temperature | 1.0 |
| Rollout cut-off length | 8192 |

### B.4 DATASET

**Cold-start Phase** For SFT cold-start, we subsample from the Openr1-Math-46k-8192 (Hugging Face, 2025) dataset. Specifically:

- Experiments in Table 1 and Table 3: We uniformly subsample 1k, 3k, and 6k splits from the verifiable questions, each paired with one corresponding demonstration. The same subsampled datasets are used to conduct SFT cold-start experiments with our proposed AESL and baseline methods.

- Experiments in Table 4: We begin by uniformly subsampling a 6k split from the original dataset. Next, the base model (Qwen2.5-7B-Instruct) is used to rank the questions by difficulty, determined by the reversed average accuracy of 8 rollouts per question. Using this ranking, we construct two splits: (i) A hard split consisting of the top 3k most difficult questions (i.e., those with the lowest accuracy). (ii) A simple split, comprising the remaining questions. The SFT cold-start experiments with our proposed AESL and baseline methods are then conducted on these two splits.

**Evaluation** For evaluation, the datasets used are listed in Table 11. Given the large evaluation variance induced by the small sizes of the AIME24, AIME25, and AMC23 datasets, we report the avg@64 metric for these datasets and pass@1 for others.

### B.5 INFRASTRUCTURE

The experiments were conducted using NVIDIA H100 Tensor Core GPUs.

Table 10: Test-time parameters used for evaluation.

| Hyperparameter | Value |
|---|---|
| Output length | 8192 |
| Temperature | 0.6 |
| Top-p | 0.95 |

Table 11: Evaluation benchmarks.

| Dataset | Number of Questions |
|---|---|
| AIME24[1] | 30 |
| AIME25[2] | 30 |
| AMC23[3] | 30 |
| MATH-500[4] | 500 |
| Minerva Math[5] | 272 |
| OlympiadBench[6] | 675 |

[1] https://huggingface.co/datasets/math-ai/aime24.
[2] https://huggingface.co/datasets/math-ai/aime25.
[3] https://huggingface.co/datasets/math-ai/amc23.
[4] https://huggingface.co/datasets/HuggingFaceH4/MATH-500.
[5] https://huggingface.co/datasets/math-ai/minervamath.
[6] https://huggingface.co/datasets/Hothan/OlympiadBench.

## C  DETAILED EXPERIMENTAL RESULTS

### C.1  ELABORATED RESULTS OF THE MOTIVATION EXAMPLE

**Detailed Evaluation Results**  We present the breakdown of results from Figure 1b across different evaluation datasets in Table 12.

Table 12: Post-training performance (↑) with different cold-start training budgets with standard CE loss.

| | AIME24 | AIME25 | AMC23 | MATH. | Min. | Olym. | Avg. |
|---|---|---|---|---|---|---|---|
| Base | 11.93 | 8.39 | 53.16 | 78.2 | 36.76 | 40.0 | 38.07 |
| +RL | 13.7 | 5.73 | **54.45** | 76.4 | 38.24 | 40.0 | 38.09 |
| +SFT (100 step) | 9.74 | 12.5 | 43.32 | 71.2 | 31.25 | 31.7 | 33.28 |
| +RL | **18.23** | 15.94 | 54.37 | **80.4** | 37.5 | **42.81** | **41.54** |
| +SFT (200 step) | 11.82 | 14.11 | 45.04 | 72.8 | 33.09 | 33.19 | 35.01 |
| +RL | 16.72 | 15.57 | 55.12 | 78.8 | 38.97 | 42.07 | 41.21 |
| +SFT (300 step) | 12.76 | 15.1 | 44.96 | 72.4 | 34.56 | 33.63 | 35.57 |
| +RL | 18.12 | **15.99** | 53.24 | 80.2 | 36.76 | 40.0 | 40.72 |
| +SFT (1 epoch) | 12.29 | 15.31 | 44.22 | 74.2 | **39.34** | 35.26 | 36.77 |
| +RL | 15.99 | 15.52 | 53.36 | 78.4 | 37.13 | 40.74 | 40.19 |

**Response Length Analysis**  We provide the average response lengths before and after RL for Qwen2.5-7B-Instruct and Qwen2.5-Math-7B in Table 13.

Overall, the average response length before RL for Qwen2.5-Math-7B is slightly longer than that of Qwen2.5-7B-Instruct. Additionally, the post-RL growth in average length for Qwen2.5-Math-7B is much higher. These results indicate that Qwen2.5-Math-7B demonstrates a superior ability to scale response length and discover long-CoT patterns autonomously, whereas Qwen2.5-7B-Instruct fails to do so through direct RL. This highlights the necessity of a pre-RL cold-start SFT phase for Qwen2.5-7B-Instruct to inject long-CoT patterns for better RL scaling.

Table 13: Averaged response length before and after RL.

|  | AIME24 | AIME25 | AMC23 | MATH. | Min. | Olym. | Avg. |
|---|---|---|---|---|---|---|---|
| Qwen2.5-7B-Instruct | 1224 | 1039 | 917 | 625 | 688 | 938 | 905 |
| +RL | 1072 | 1061 | 1046 | 712 | 721 | 1180 | 965 |
| Qwen2.5-Math-7B | 1487 | 1517 | 1016 | 808 | 691 | 1074 | 1099 |
| +RL | 2689 | 2317 | 1445 | 891 | 935 | 1550 | 1638 |

## C.2 OTHER PERFORMANCE-BASED EVALUATION METRICS

In addition to accuracy, as discussed in Figure 1b, we evaluate the pass@8 metric during the cold-start phase to assess its suitability as an early-stopping criterion. To ensure the robustness of the estimated pass@8 metric, we use 16 rollouts per question, which provides a more reliable measure of model performance by reducing the variance of the metric across questions. The results, shown in Figure 4, reveal that pass@8 follows a similar trend to evaluation accuracy, reflecting the characteristic shift-and-readaptation process observed during cold-start training.

However, despite its alignment with evaluation accuracy, pass@8 fails to reliably predict RL potential during the lightweight cold-start phase. This indicates that pass@8, like accuracy, is an inadequate choice for guiding early stopping in this context, as it does not effectively capture the diversity dynamics necessary for effective RL preparation.

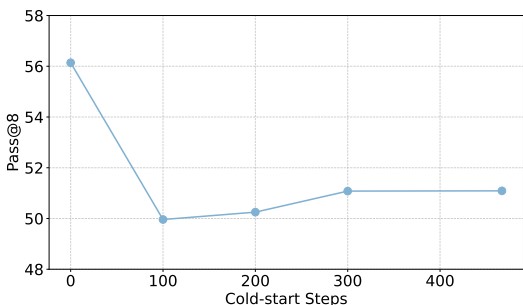

Figure 4: The Pass@8 metric throughout the cold-start training process, corresponding to Figure 1b. The trend mirrors evaluation accuracy, demonstrating the shift-and-readaptation process.

## C.3 EXTENDED ANALYSIS ON EXPERIMENTS

To further investigate the mechanism of AESL, we analyze the word clouds generated after the cold-start phase using CE loss and AESL. From Figure 5, we observe that both methods learn the new reasoning pattern effectively, as evidenced by the increased frequency of branching words (Wang et al., 2025) (e.g., "wait" and "maybe" ) aligned with the dataset. However, the CE loss-based cold-start exhibits more severe over-memorization of specific dataset elements, such as repeated high-frequency numbers (e.g., "12"). In contrast, AESL mitigates this issue by striking a balance between learning the new reasoning pattern and preserving the base model's original distribution.

In addition to the word cloud analysis, the results in Table 3 and Table 4 demonstrate that AESL consistently outperforms traditional CE-based cold-start methods across varying dataset sizes and difficulty levels. By comparing the results across different cold-start dataset sizes in Table 3, it is evident that scaling the dataset size during the cold-start phase improves performance. Furthermore, comparing the results in Table 4 and Table 1 reveals that using a mixture of difficulty levels provides superior RL preparation compared to relying solely on either the simple or hard splits of the dataset.

These findings suggest a promising direction for future research: optimizing the curation of cold-start datasets to better prepare LLMs for subsequent RL training. Such dataset curation, combined with improved loss functions like AESL, could further enhance the base model's capacity to balance learning new reasoning patterns while preserving its original distribution, ultimately improving its readiness for downstream RL training.

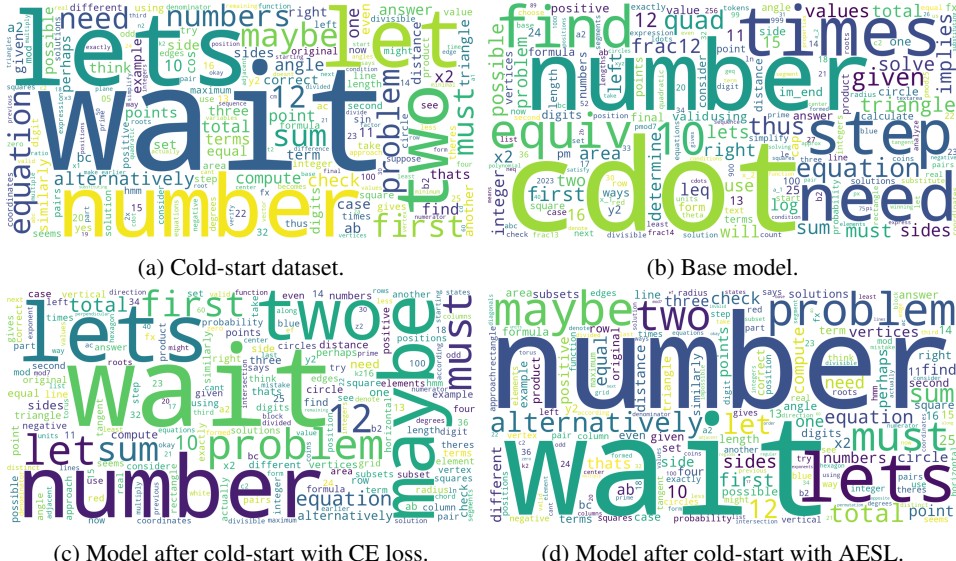

(a) Cold-start dataset.

(b) Base model.

(c) Model after cold-start with CE loss.

(d) Model after cold-start with AESL.

Figure 5: Word frequency analysis showing vocabulary patterns across different methods.

## C.4 Computational Overhead

While AESL introduces improved preparation of LLMs prior to the RL phase, its more complex form compared to standard CE loss inevitably introduces additional overhead. In our implementation, we trade off memory complexity for time complexity, leading to approximately 20% more training time compared to the standard CE loss.

However, it is important to note that, this work focuses on the cold-start phase, and the proposed loss is specifically designed for this stage. The cold-start phase is typically far less resource-intensive compared to the subsequent RL phase. For example, in our experimental settings, the RL phase requires approximately $10\times$ more wall-clock time than the cold-start phase when using the same computational resources. This is primarily due to the larger dataset size and slower convergence during RL.

As a result, the additional time complexity introduced by AESL in the cold-start phase accounts for only 2% of the total post-training time. We believe this trade-off is worthwhile given the improved performance it provides.

## D Extended Experiments

To validate our insights beyond mathematical tasks, we conducted additional experiments on general reasoning benchmarks.

### D.1 Setup

**Base Model and Algorithm** We inherit the environmental settings described in Section 4 and use Qwen2.5-7B-Instruct as the base model alongside the GRPO algorithm for RLVR. Math-Verify is employed as the verifier.

**Datasets** For SFT cold-start training, we subsample data from the OpenThoughts-114k (Guha et al., 2025) dataset to provide general reasoning demonstrations and combine it with subsampled data from Openr1-Math-46k-8192 (Hugging Face, 2025) for format alignment. Specifically, we filtered out coding tasks as our initial experiments revealed that their inclusion (due to their non-boxed output format) significantly hindered the model's ability to generate boxed answers during general reasoning tasks. From these filtered results, we uniformly subsampled 3k examples and mixed them with 3k examples uniformly subsampled from Openr1-Math-46k-8192.

For RL post-training, we subsampled 20k examples from the general reasoning dataset WebInstruct-verified (Ma et al., 2025), which contains general reasoning questions paired with verifiable answers.

For evaluation, the datasets used are summarized in Table 14.

Table 14: Evaluation benchmarks for general reasoning.

| Dataset | Number of Questions |
|---|---|
| ARC-C[1] | 1172 |
| GPQA-Dimond[2] | 198 |
| MMLU-Pro[3] | 12032 |

[1] `https://huggingface.co/datasets/allenai/ai2_arc`.
[2] `https://huggingface.co/datasets/fingertap/GPQA-Diamond`.
[3] `https://huggingface.co/datasets/TIGER-Lab/MMLU-Pro`.

## D.2 RESULTS

The results are presented in Table 15. We observe that the proposed AESL method consistently outperforms the baseline method using CE-loss. These findings confirm the effectiveness of AESL beyond the mathematical domain and emphasize the importance of dynamically balancing new pattern acquisition with the preservation of the base model's original distribution during the post-train cold-start phase.

Table 15: Post-training performance (↑) with different cold-start methods. Avg. denotes the macro-average across benchmarks.

| | ARC-C | GPQA-Dimond | MMLU-Pro | Avg. |
|---|---|---|---|---|
| Base | 87.97 | 19.19 | 53.15 | 47.44 |
| +SFT (CE-(ES)) | 52.22 | 16.67 | 42.48 | 37.12 |
| +RL | 87.88 | 19.19 | 48.51 | 51.86 |
| +SFT (AESL) | 68.27 | 17.17 | 43.17 | 42.84 |
| +RL | **90.1** | **32.32** | **54.21** | **58.88** |

* CE and CE-ES checkpoints coincide, so results are reported together.

## E THE USE OF LLMS

LLMs are used to polish the writing in this paper.

