# OpenReview forum: "Getting Your LLMs Ready for Reinforcement Learning with Lightweight SFT"
_ICLR.cc/2026/Conference — ICLR 2026 Poster_

### Official Review · Reviewer_Wffo · 2025-10-30

**Soundness:** 2
**Presentation:** 3
**Contribution:** 2
**Rating:** 2
**Confidence:** 4

**Summary:**

In this paper, the authors show that when doing SFT before RL, the checkpoint with the best validation performance often isn't the best starting point for RL. The authors find this happens because models drift too far from the base distribution before traditional overfitting occurs. They propose two solutions: (1) using diversity metrics (entropy, self-BLEU) for checkpoint selection instead of validation performance, and (2) Adaptive Early-Stop Loss that balances learning new patterns while preserving the base model's distribution.

**Strengths:**

1. The observation that best SFT performance does not translate to best RL readiness is valuable and practically very relevant.
2. The paper is well-motivated and easy to follow.

**Weaknesses:**

1. I feel that in this paper, severely limited scope threatens generalizability of the results. For example, only mathematical reasoning tasks tested that too only on Qwen model family (both 7B variants are closely related). There's no evidence this generalizes to code, instruction-following, creative writing, etc. or is applicable to a different model family like Llama 3 8B.

2. The AESL weighting function in equation 4 appears ad-hoc. Why softmax? Why this specific ratio? Why average log-probability in the denominator? The entropy decomposition (appendix A) shows a decomposition exists but doesn't strongly motivate this particular functional form. Moreover, there's no comparison to simpler alternatives (linear weighting by confidence, threshold-based stopping loss, etc.) An ablation comparing 3-4 different weighting schemes would greatly strengthen the paper.

3. [Minor] There's no significance testing despite small margins (many improvements are 1-2 points in Tables 1-4).

4. AESL requires additional forward passes for computing weights. How much does this add to training time? Is the improvement worth the computational overhead?


Overall, in its current form, this paper feels more like a useful engineering trick for a specific setting (math reasoning with Qwen) rather than a fundamental contribution to post-training research. I'd encourage the authors to either (a) substantially expand the scope to other domains to demonstrate generality, or (b) provide much deeper analysis of the specific math/Qwen setting to get broader insights.
Right now, the contribution in this paper feels incremental for ICLR but with significant additional work, this could become a solid accept.

**Questions:**

1. I am curious about under what conditions does AESL underperform standard approaches? Are there cases where preserving base distribution hurts?
2. How does AESL compare to simply doing less SFT (fewer epochs)? Is the sophisticated weighting necessary or would early stopping at lower loss achieve similar results?
3. Is this approach applicable when use a model which hasn't been RL'd before? Qwen-Math already has been RL'd before. For example, Olmo-2 provides checkpoints right after pre-training.

---

> ### Author Response · Authors · 2025-11-30
> **Rebuttal (1/2)**
>
> We sincerely appreciate the reviewer’s constructive feedback and insightful suggestions. Your comments have helped us refine our work and address several key aspects. Below, we provide detailed responses to your points and summarize the updates made to the manuscript.
>
> **Q1:** severely limited scope threatens generalizability of the results
>
> **A1:** Thank you for pointing this out. Following your suggestion, we have extended our experiments to broaden the scope of our evaluation in two key directions:
>
> - Base Models: We now include results using Llama-3.1-8B-Instruct as the base model (reported in Table 5 of the updated manuscript).
> - Datasets: Beyond mathematical tasks, we have expanded our experiments to include general reasoning tasks (results in Table 15, with experimental details in Appendix D).
>
> Below, we summarize the key findings from these newly added experiments (full results are in the updated manuscript):
>
> **Table:** Post-training performance ($\uparrow$) of different cold-start methods using Llama-3.1-8B-Instruct as the base model.
>
> | Method            | AIME24 | AIME25 | AMC23  | MATH.  | Min.   | Olym.  | Avg.   |
> |-------------------|--------|--------|--------|--------|--------|--------|--------|
> | **Base**              | 5.26   | 0.68   | 19.14  | 45.6   | 23.16  | 16.89  | 18.46  |
> | **+SFT (CE) + RL**| 4.17   | 2.45   | 24.57  | 58.0   | 24.26  | 21.33  | 22.46  |
> | **+SFT (CE-ES) + RL** | 5.36   | 1.93   | 28.12  | 59.8   | **29.04**| 22.22 | 24.41  |
> | **+SFT (AESL) + RL** | **6.56** | **3.8** | **31.21** | **60.4** | 28.68 | **22.37** | **25.5** |
>
>
> **Table:** Post-training performance ($\uparrow$) with different cold-start methods using general reasoning dataset.
>
> | Method              | ARC-C   | GPQA-Diamond | MMLU-Pro | Avg.   |
> |---------------------|---------|--------------|----------|--------|
> | Base                | 87.97   | 19.19        | 53.15    | 47.44  |
> | **+SFT (CE-(ES)) + RL** | 87.88   | 19.19        | 48.51    | 51.86  |
> | **+SFT (AESL) + RL**| **90.1**| **32.32**    | **54.21**| **58.88** |
>
>
> Overall, we see that AESL consistently outperforms baseline methods across all experiments. These results strongly support the generalizability of our findings across models and tasks.
>
> With the inclusion of newly added experiments, **our evaluation now spans three distinct base models from two model families, two diverse tasks (mathematics and general reasoning), and multiple cold-start dataset configurations that vary in size and difficulty.** This extensive experimental setup ensures that our findings are robust, comprehensive, and applicable across a wide range of scenarios.
>
> **Q2:** An ablation comparing 3-4 different weighting schemes would greatly strengthen the paper
>
> **A2:** We appreciate the suggestion, and we have included several ablations with alternative designs. The results are summarized below (full table and detailed descriptions can be found in Table 6 of the updated manuscript):
>
> **Table:** Ablation studies for AESL
>
> | Method                  | Performance (Avg.) |
> |-------------------------|--------------------|
> | Base                   | 38.09             |
> | AESL                   | **42.26**            |
> | AESL with constant denominator      | 40.35             |
> | AESL with ranking-based threshold   | 40.6              |
> | AESL w/ linear weighting for prefix confidence  | 40.96             |
>
> The results show that all alternative designs lead to performance drops compared to AESL, underscoring the effectiveness of AESL’s design. As discussed in Appendix A, the inclusion of prefix confidence is motivated by the observation that different tokens require varying levels of balance between learning from the new dataset and preserving the original distribution. The averaging mechanism is implemented as the multiplication inside log function, with a normalization factor based on the prefix length. Additionally, the use of the softmax function in our method is empirically supported by insights gained during our initial experiments.
>
> **Q3:** no significance testing
>
> **A3:** Our experiment settings are consistent with standard practices in the field [1–3]. Due to the computational cost of RL post-training, conducting multiple runs with different seeds is infeasible. However, the superior performance of AESL across the six math evaluation benchmarks and the extensive range of experiments (see A2) strongly supports the robustness of our findings.

---

> ### Author Response · Authors · 2025-11-30
> **Rebuttal (2/2)**
>
> **Q4:** additional computational overhead
>
> **A4:** Thank you for the question. The proposed loss does require additional 20% more training time compared to standard CE loss for the SFT.
>
> However, we would like to emphasize that, as indicated by the title of our paper, **this work focuses on the cold-start phase**, and the proposed loss is specifically designed for this phase. The cold-start phase is typically far less resource-intensive compared to the subsequent RL phase. For instance, in our experimental settings, the RL phase requires 10× more wall-clock time than the cold-start phase when using the same computational resources, due to a larger dataset and slower convergence. **As a result, the additional time complexity introduced by AESL in the cold-start phase accounts for only 2% of the total post-training time. Given the performance improvements achieved with AESL, we believe this trade-off is well-justified.**
>
> We have now included this discussion in Appendix C.4 of the updated manuscript.
>
> **Q5:** I am curious about under what conditions does AESL underperform standard approaches? Are there cases where preserving base distribution hurts?
>
> **A5:** AESL is specifically designed for the cold-start phase, where the dataset is small, and overfitting is a major concern. In contrast, if the dataset is large and of high quality, techniques like AESL are unnecessary since the model can safely fit the new distribution without significant risk of overfitting. In such cases, preserving the base model distribution may even hinder performance.
>
>
> **Q6:** How does AESL compare to simply doing less SFT (fewer epochs)? Is the sophisticated weighting necessary or would early stopping at lower loss achieve similar results?
>
> **A6:** We have shown that AESL outperforms "simply doing less SFT" and that "doing less SFT" outperforms "SFT until performance drops." **We agree that "simply doing less SFT" is an important baseline, which is why we included it (denoted as CE-ES, where ES stands for Early Stopping) in all our experiments across Tables 1, 3, 4, 5, and 15 in the original manuscript.**
>
> In conclusion, we are deeply grateful for the reviewer’s thoughtful feedback and constructive suggestions. The additional experiments, ablation studies, and clarifications have significantly strengthened our manuscript. These updates demonstrate the robustness, generalizability, and practical relevance of AESL across diverse tasks and settings. Thank you again for your time and effort in helping us improve this work.
>
> -------
> [1] "Preserving diversity in supervised fine-tuning of large language models", ICLR 2025 \
> [2] "Open r1: A fully open reproduction of deepseek-r1", Blog \
> [3] "Understanding R1-Zero-Like Training: A Critical Perspective", COLM 2025

---

### Official Review · Reviewer_H9W3 · 2025-10-31

**Soundness:** 3
**Presentation:** 3
**Contribution:** 3
**Rating:** 6
**Confidence:** 3

**Summary:**

The paper finds SFT checkpoints with best evaluation performance fail for RL due to distributional forgetting. Diversity metrics are better early-stopping criteria. Furthermore, it proposes AESL, balancing new pattern learning and base distribution preservation, outperforming baselines in math reasoning tasks.

**Strengths:**

(1) It pinpointed the core misalignment between traditional SFT and RL preparation, uncovering "distributional forgetting"
(2) Developed AESL with token/subsequence-level adaptive weighting.
(3) Conducted in-depth mechanism analysis and reported multiple metrics, enhancing result interpretability and credibility.

**Weaknesses:**

(1) While the paper identifies "distributional forgetting" as a key bottleneck, it lacks a deep investigation into its underlying causes.
(2) It remains unproven whether AESL’s effectiveness translates to other domains (e.g., code generation, scientific writing).

**Questions:**

(1) Could you provide experimental evidence to verify whether AESL’s mechanism of balancing new pattern learning and base distribution preservation generalizes to other domains where RL post-training for LLMs is critical?
(2) Could you conduct layer-wise parameter analysis or data-specific ablation studies to discuss the root mechanisms of distributional forgetting?

---

> ### Author Response · Authors · 2025-11-30
> **Rebuttal**
>
> We sincerely appreciate the reviewer’s positive feedback and thoughtful suggestions. We are glad to see the reviewer recognize the significance of our work. In response to the questions and recommendations, we have carefully addressed them and made corresponding updates to the manuscript. Below, we provide detailed clarifications and a summary of the revisions.
>
> **Q1:** other domains
>
> **A1:** Thank you for the suggestion. We have expanded our experiments beyond mathematical tasks to include general reasoning tasks. The results, along with experimental details, are presented in Table 15 and Appendix D of the updated manuscript. Below, we summarize the key findings:
>
> **Table:** Post-training performance ($\uparrow$) with different cold-start methods using general reasoning dataset.
>
> | Method              | ARC-C   | GPQA-Diamond | MMLU-Pro | Avg.   |
> |---------------------|---------|--------------|----------|--------|
> | Base                | 87.97   | 19.19        | 53.15    | 47.44  |
> | **+SFT (CE-(ES)) + RL** | 87.88   | 19.19        | 48.51    | 51.86  |
> | **+SFT (AESL) + RL**| **90.1**| **32.32**    | **54.21**| **58.88** |
>
>
> Overall, we see that AESL consistently outperforms baseline methods on general reasoning tasks, further validating its effectiveness. With the newly added experiments, **our evaluation now spans three distinct base models from two model families, two diverse tasks (mathematics and general reasoning), and multiple cold-start dataset configurations that vary in size and difficulty.** This extensive experimental setup demonstrates the robustness, comprehensiveness, and generalizability of AESL across a wide range of scenarios.
>
>
> **Q2:** distributional forgetting
>
> **A2:** As discussed in our manuscript, such forgetting likely stems from overfitting to the small cold-start dataset. This observation is supported by experimental evidence provided in Figures 1 & 2 and Table 2, which show a clear trend during cold-start training: while there is an increase in performance post-SFT, this is accompanied by a decrease in diversity and RL potential. These trends suggest that the model begins to rely heavily on copying answers from the cold-start demonstrations, potentially losing some of the broader knowledge it had acquired during earlier training phases. This phenomenon highlights the trade-offs inherent in the cold-start phase. Successfully balancing the preservation of the base model’s existing knowledge with the need to adapt to new patterns introduced during cold-start training is crucial.
>
> We are grateful to the reviewer for the thoughtful feedback, which has allowed us to refine and strengthen our manuscript further. The inclusion of additional experiments and analyses has enhanced the robustness and generalizability of our evaluation. We sincerely appreciate the reviewer’s time and effort in helping us improve this work.

---

### Official Review · Reviewer_iMAH · 2025-11-01

**Soundness:** 3
**Presentation:** 3
**Contribution:** 3
**Rating:** 6
**Confidence:** 4

**Summary:**

This paper explores the best strategy for fine-tuning a base LLM checkpoint in order to achieve the best performance after RL fine-tuning. Currently, when RL fine-tuning is used to improve a reasoning LLM's ability to solve math or coding problems, a short supervised fine-tuning (SFT) is done on a base checkpoint. One reason for this, as mentioned in the paper, is to provide the base LLM with useful reasoning patterns (that the base model does not have, or is quite weak on), which are further enforced via RL.

A key finding of the paper is that the commonly used metrics such as accuracy on downstream task during SFT does not correspond to the best performance after RL fine-tuning. In fact, the paper shows that lower accuracies during SFT (somewhat surprisingly) leads to the best performance after RL fine-tuning. The paper then proposes to use diversity metrics such as entropy or self-BLEU scores to pick the best checkpoints during the SFT stage, in order to maximize performance after RL.

The paper also proposes a new method of doing SFT before RL fine-tuning that, by using an adaptive weighting on the loss function, maintains diversity of the SFT model while also learning the necessary reasoning patterns for RL. Experiments show that the method improves over existing SFT + RL baselines on math benchmarks, for the Qwen class of models.

**Strengths:**

1. Empirically showing that commonly used metrics for SFT are not optimal for RL fine-tuning. This is an interesting finding that is somewhat counter to existing conventional wisdom.

2. Proposed new methods for SFT fine-tuning target. The paper proposes to use diversity metrics such as entropy or self-BLEU to select checkpoints during SFT before RL fine-tuning. This is a well-motivated idea (maintain diversity to avoid model collapse and allow more exploration during RL) and seems to be supported by empirical evidence.

3. Proposed a change to the standard cross-entropy loss during SFT in order to optimize for diversity metrics while allowing the model to learn new useful reasoning patterns.

4. Experimental evidence for the effectiveness of the proposed method, and ablations.

**Weaknesses:**

1. Lacks analysis of the complexity of the proposed loss. How much additional complexity does the weighting require? Are there significant memory requirements to handle the logits for each training data sequence?

2. Lack of diversity in experiments. Only the Qwen class of models is used in experiments. Why are there no other open-source, small-sized models? The evaluations are also only done on math benchmarks. It is difficult to be convinced of the generality of the proposed method when the breadth of results are quite limited.

3. Lack of details of baselines. What are GEM and PSFT? You especially mention that GEM "emphasizes promoting diversity" and that PSFT is "concurrent work". How similar are they to the method proposed in this paper?

**Questions:**

Appendix A, equation (7)-(8), what are the \pi(s_t|s_{t-1})'s, are they deltas functions?

Page 5, line 260, what does the index j index over?

Table 1, why are there no error intervals? How many seeds were used?

---

> ### Author Response · Authors · 2025-11-30
> **Rebuttal (1/2)**
>
> Thank you for your positive review and valuable feedback. We are pleased that the reviewer found our work interesting and appreciate the suggestions. In response, we have carefully addressed your comments, incorporated the necessary revisions into the manuscript, and summarize our updates and clarifications below.
>
> **Q1:** Lacks analysis of the complexity of the proposed loss.
>
> **A1:** Thank you for raising this point. The proposed loss does require additional memory to handle the logits. In our implementation, we trade off memory complexity for time complexity, resulting in approximately 20% more training time compared to standard CE loss.
>
> However, we would like to emphasize that, as indicated by the title of our paper, **this work focuses on the cold-start phase**, and the proposed loss is specifically designed for this phase. The cold-start phase is typically far less resource-intensive compared to the subsequent RL phase. For instance, in our experimental settings, the RL phase requires 10× more wall-clock time than the cold-start phase when using the same computational resources, due to a larger dataset and slower convergence. **As a result, the additional time complexity introduced by AESL in the cold-start phase accounts for only 2% of the total post-training time. Given the performance improvements achieved with AESL, we believe this trade-off is well-justified.**
>
> We have now included this discussion in Appendix C.4 of the updated manuscript.
>
> **Q2:** Lack of diversity in experiments.
>
> **A2:** We acknowledge the reviewer’s concern regarding the diversity of the experiments. Following your suggestion, we have extended our experiments in two key directions:
> - Base Models: We now include results using Llama-3.1-8B-Instruct as the base model (reported in Table 5 of the updated manuscript).
> - Datasets: We have broadened the scope beyond mathematical tasks to general reasoning tasks (results in Table 15, with experimental details in Appendix D).
>
> Below, we summarize the key findings from these newly added experiments (full results are in the updated manuscript):
>
> **Table:** Post-training performance ($\uparrow$) of different cold-start methods using Llama-3.1-8B-Instruct as the base model.
>
> | Method            | AIME24 | AIME25 | AMC23  | MATH.  | Min.   | Olym.  | Avg.   |
> |-------------------|--------|--------|--------|--------|--------|--------|--------|
> | **Base**              | 5.26   | 0.68   | 19.14  | 45.6   | 23.16  | 16.89  | 18.46  |
> | **+SFT (CE) + RL**| 4.17   | 2.45   | 24.57  | 58.0   | 24.26  | 21.33  | 22.46  |
> | **+SFT (CE-ES) + RL** | 5.36   | 1.93   | 28.12  | 59.8   | **29.04**| 22.22 | 24.41  |
> | **+SFT (AESL) + RL** | **6.56** | **3.8** | **31.21** | **60.4** | 28.68 | **22.37** | **25.5** |
>
>
> **Table:** Post-training performance ($\uparrow$) with different cold-start methods using general reasoning dataset.
>
> | Method              | ARC-C   | GPQA-Diamond | MMLU-Pro | Avg.   |
> |---------------------|---------|--------------|----------|--------|
> | Base                | 87.97   | 19.19        | 53.15    | 47.44  |
> | **+SFT (CE-(ES)) + RL** | 87.88   | 19.19        | 48.51    | 51.86  |
> | **+SFT (AESL) + RL**| **90.1**| **32.32**    | **54.21**| **58.88** |
>
>
> Overall, we see that AESL consistently outperforms baseline methods across all experiments. With the inclusion of newly added experiments, **our evaluation now spans three distinct base models from two model families, two diverse tasks (mathematics and general reasoning), and multiple cold-start dataset configurations that vary in size and difficulty.** This extensive experimental setup demonstrates the robustness, comprehensiveness, and generalizability of AESL across a wide range of scenarios.
>
> **Q3:** Lack of details of baselines.
>
> **A3:** We would like to provide additional clarifications:
> - Methods: GEM and PSFT propose new training objectives for the SFT post-training phase. Unlike AESL, these baselines focus solely on token-level objectives and do not incorporate sub-sequence information as AESL does.
> - Application Scenarios: AESL is specifically tailored for the lightweight cold-start phase, which prepares the model for the subsequent RL phase, whereas GEM and PSFT are designed for scenarios with abundant demonstration data available for large-scale SFT.

---

> ### Author Response · Authors · 2025-11-30
> **Rebuttal (2/2)**
>
> **Q4:** Appendix A, equation (7)-(8), what are the \pi(s_t|s_{t-1})'s, are they deltas functions?
>
> **A4:** No, $\pi$ refers to the policy model (LLM).
>
> **Q5:** Page 5, line 260, what does the index j index over?
>
> **A5:** It is index over the possible tokens in the vocabulary.
>
> **Q6:** Table 1, why are there no error intervals? How many seeds were used?
>
> **A6:** We used one seed (42) for all experiments, consistent with standard practices in the field [1–3]. Due to the computational cost of RL post-training, conducting multiple runs with different seeds is infeasible. However, the superior performance of AESL across the six math evaluation benchmarks and the extensive range of experiments (see A2) strongly supports the robustness of our findings.
>
> We sincerely thank the reviewer for their feedback, which has significantly enhanced the quality of our manuscript. The new experiments further validate the robustness and generalizability of AESL. We appreciate the time and effort the reviewer has dedicated to improving our work.
>
>
> -------
> [1] "Preserving diversity in supervised fine-tuning of large language models", ICLR 2025 \
> [2] "Open r1: A fully open reproduction of deepseek-r1", Blog \
> [3] "Understanding R1-Zero-Like Training: A Critical Perspective", COLM 2025

---

### Official Review · Reviewer_WyUj · 2025-11-02

**Soundness:** 2
**Presentation:** 2
**Contribution:** 2
**Rating:** 4
**Confidence:** 3

**Summary:**

Reinforcement learning fine-tuning of LLMs typically involves a SFT phase that primes the model for more efficient RL training. In this context, this paper studies the question of how to select the right SFT checkpoint to initialize RL training from. The authors claim that diversity metrics (entropy, self-BLEU) are better correlated with final RL performance than standard CE loss, and propose an adaptive early-stop loss (AESL) for SFT that consists of weighted CE where the weight is designed to scale decreasingly with both the likelihood of the target token and the likelihood of the prefix sequence.

**Strengths:**

This paper studies the important yet under-studied question of how SFT checkpoint selection influences final RL performance. This is especially critical because all modern LLM training pipelines involve SFT and RL phases for nearly all tasks, including reasoning and human alignment. The paper is clearly written and the proposed method is easy to understand.

**Weaknesses:**

I found the empirical motivation and evidence to be a bit lacking. In section 3, the authors claim that diversity has a clear correlation with final performance, but only present results for two (related) models and a single SFT dataset. More broadly showing that this is the case on more models (especially non-Qwen ones), different datasets, and different tasks (e.g. RLHF) would immensely strengthen the claims in the paper.

I am also not strongly convinced by the argument presented in Figure 3, and would also like to see an ablation of the denominator in the loss weight.

**Questions:**

1. Cold start seems like the wrong term to use here - warm-start is more appropriate. I know Deepseek-R1 paper also uses this term improperly, but it should be corrected here.
2. Table 5 should include vanilla SFT as well. Along with varying t_scaling, I would also like to see a sweep with the average prefix log prob term removed.

---

> ### Author Response · Authors · 2025-11-30
> **Rebuttal (1/2)**
>
> Thank you for your thoughtful review and constructive feedback. We greatly appreciate the reviewer’s recognition of the significance of our research question and the clarity of our writing. In response, we have carefully incorporated your suggestions into the revised manuscript and would like to summarize the updates and provide clarifications below.
>
> **Q1:** More empirical evidence.
>
> **A1:** Thank you for the suggestion.
> 1. **Clarification of Current Evidence**: We would like to clarify that the correlation between post-cold-start diversity and final performance has been extensively validated through multiple experiments, as shown in Figure 1 and Table 2. The SFT datasets used for these experiments are sampled differently from the same superset, varying in dataset size and difficulty to synthesize different cold-start settings. These experiments were deliberately designed to investigate how dataset properties influence performance. Beyond the main results in Table 1, additional analyses on cold-start dataset size and difficulty are provided in Tables 3 and 4, which consistently demonstrate the superiority of AESL over baseline methods.
> 2. **New Experiments**: We acknowledge the reviewer’s concern regarding the diversity of the experiments. Following your suggestion, we have extended our experiments in two key directions:
>     - Base Models: We now include results using Llama-3.1-8B-Instruct as the base model (reported in Table 5 of the updated manuscript).
>     - Datasets: We have broadened the scope beyond mathematical tasks to general reasoning tasks (results in Table 15, with experimental details in Appendix D).
>
>     Below, we summarize the key findings from these newly added experiments (full results are in the updated manuscript):
>
>     **Table:** Post-training performance ($\uparrow$) of different cold-start methods using Llama-3.1-8B-Instruct as the base model.
>
>     | Method            | AIME24 | AIME25 | AMC23  | MATH.  | Min.   | Olym.  | Avg.   |
>     |-------------------|--------|--------|--------|--------|--------|--------|--------|
>     | **Base**              | 5.26   | 0.68   | 19.14  | 45.6   | 23.16  | 16.89  | 18.46  |
>     | **+SFT (CE) + RL**| 4.17   | 2.45   | 24.57  | 58.0   | 24.26  | 21.33  | 22.46  |
>     | **+SFT (CE-ES) + RL** | 5.36   | 1.93   | 28.12  | 59.8   | **29.04**| 22.22 | 24.41  |
>     | **+SFT (AESL) + RL** | **6.56** | **3.8** | **31.21** | **60.4** | 28.68 | **22.37** | **25.5** |
>
>
>     **Table:** Post-training performance ($\uparrow$) with different cold-start methods using general reasoning dataset.
>
>     | Method              | ARC-C   | GPQA-Diamond | MMLU-Pro | Avg.   |
>     |---------------------|---------|--------------|----------|--------|
>     | Base                | 87.97   | 19.19        | 53.15    | 47.44  |
>     | **+SFT (CE-(ES)) + RL** | 87.88   | 19.19        | 48.51    | 51.86  |
>     | **+SFT (AESL) + RL**| **90.1**| **32.32**    | **54.21**| **58.88** |
>
>
>     Overall, we see that AESL consistently outperforms baseline methods across all experiments. The correlation between post-cold-start diversity and final performance is validated: SFT (AESL) > SFT (CE-ES) > SFT (CE).
>     These results strongly support the generalizability of our findings across models and tasks.
>
> With the inclusion of newly added experiments, **our evaluation now spans three distinct base models from two model families, two diverse tasks (mathematics and general reasoning), and multiple cold-start dataset configurations that vary in size and difficulty.** This extensive experimental setup ensures that our findings are robust, comprehensive, and applicable across a wide range of scenarios.
>
> **Q2:** Cold start seems like the wrong term to use here - warm-start is more appropriate. I know Deepseek-R1 paper also uses this term improperly, but it should be corrected here.
>
> **A2:** This is an interesting point. In fact, we can see both sides:
> - Warm-Start: From the purpose of this phase, "warm-start" is more appropriate as it aims to prepare the base model for subsequent RL fine-tuning by introducing reasoning patterns (e.g., long CoT) and enabling faster convergence.
> - Cold-Start: The term "cold-start" likely originates from recommendation systems, referring to scenarios where limited data is available in a new domain. In our work, we adopt "cold-start" to emphasize our focus on scenarios where high-quality demonstration data are limited. During this phase, we employ lightweight SFT with a small, high-quality dataset to steer the model to a better RL starting point.
>
> While "warm-start'" better conveys the purpose, "cold-start" better reflects the focus of this paper on addressing challenges in data-scarce setups. Furthermore, we have chosen to retain "cold-start" to align with prior works and ensure consistency for readers familiar with this terminology.

---

> ### Author Response · Authors · 2025-11-30
> **Rebuttal (2/2)**
>
> **Q3:** Table 5 (now Table 7) should include vanilla SFT as well
>
> **A3:** We initially excluded vanilla SFT results to keep the table concise. Following your suggestion, we have now included these results in the updated manuscript for completeness.
>
>
> **Q4:** An ablation of the denominator in the loss weight
>
> **A4:** Thank you for this suggestion. We have included an ablation study that removes the average prefix log-probability term from the denominator. Additionally, we compared several other alternative denominator designs. The results are summarized below (full table in the updated manuscript in Table 6):
>
> **Table:** Ablation studies for AESL
>
> | Method              | Performance |
> |---------------------|-------------|
> | AESL  | 42.26  |
> | AESL with the average prefix log prob term removed| 40.35 |
>
> Overall, removing the prefix log-probability term degrades performance, underscoring the importance of explicitly accounting for prefix prediction confidence in AESL. This finding aligns with our motivation discussed in Appendix A.
>
>
> We thank the reviewer for the valuable feedback, which has significantly improved the manuscript. The updates, including new experiments, improved tables, and ablation studies, demonstrate the robustness and generalizability of AESL. We hope these clarifications and updates address your concerns comprehensively.

---

### Meta-Review · Area_Chair_aPpe · 2026-01-06

**Summary:**

This submission studies an under-explored but practically important question: selecting/constructing an SFT “cold-start” checkpoint that maximizes subsequent RL performance. The paper reports that SFT checkpoints with the best validation performance can be suboptimal for RL due to early distributional drift, and proposes using diversity metrics (entropy/self-BLEU) for early stopping. It further introduces AESL, a lightweight re-weighted SFT objective that balances learning new patterns and preserving the base distribution at token and subsequence levels.
Reviewers agree the problem setting is timely and relevant to modern post-training pipelines, and the proposed AESL is simple to apply. The rebuttal substantially strengthened the empirical evidence by adding experiments on an additional model family (Llama-3.1-8B-Instruct), extending evaluation beyond math to general reasoning, and providing ablations supporting key design choices (notably the denominator/prefix-confidence term). The authors also clarified computational overhead.

**Reviewer Concerns:**

1. Statistical reliability: experiments are conducted with only a single random seed (iMAH/Wffo).

2. The experimental domains are too limited: experiments are conducted only on Qwen + math tasks; the rebuttal adds results on LLaMA + general reasoning (WyUj/iMAH/Wffo/H9W3).

3. The rationality of the ablation experiment settings (WyUj/Wffo).

4. Training overhead (iMAH/Wffo).

**Reviewer Scores:**

Reviewer WyUj may increase the score to 6, as the authors added results on LLaMA-3.1-8B-Instruct with general reasoning tasks in the rebuttal, which alleviates the core concern.

Reviewers iMAH and H9W3 are likely to maintain their scores. In the rebuttal, the authors added LLaMA-3.1-8B-Instruct + general reasoning results and quantified the overhead of the AESL SFT stage, addressing some concerns but not enough to warrant a score change.

Reviewer Wffo may slightly increase the score. Their main concerns focus on the limited scope, the ad-hoc nature of AESL, and weak novelty. Although the authors provided explanations and additional experiments in the rebuttal, it may still be difficult to fully convince this reviewer.

---

### Decision · Program_Chairs · 2026-01-26

Accept (Poster)